# Identification of Shale Lithofacies from FMI Images and ECS Logs Using Machine Learning with GLCM Features

**Min Tian** [1,2], **Maojin Tan** [1,*] and **Min Wang** [2]

1   School of Geophysics and Information Technology, China University of Geosciences (Beijing), Beijing 100083, China; tianmin502.slyt@sinopec.com
2   Research Institute of Exploration and Development, Sinopec Shengli Oilfield Company, Dongying 257015, China; wangmin136.slyt@sinopec.com
*   Correspondence: tanmj@cugb.edu.cn; Tel.: +86-0546-8715414

**Abstract:** The identification of sedimentary structures in lithofacies is of great significance to the exploration and development of Paleogene shale in the Boxing Sag. However, due to the scale mismatch between the thickness of laminae and the vertical resolution of conventional wireline logs, the conventional lithofacies division method fails to realize the accurate classification of sedimentary structures and cannot meet the needs of reservoir research. Therefore, it is necessary to establish a lithofacies identification method with higher precision from advanced logs. In this paper, a method integrating the gray level co-occurrence matrix (GLCM) and random forest (RF) algorithms is proposed to classify shale lithofacies with different sedimentary structures based on formation micro-imager (FMI) imaging logging and elemental capture spectroscopy (ECS) logging. According to the characteristics of shale laminae on FMI images, GLCM, an image texture extraction tool, is utilized to obtain texture features reflecting sedimentary structures from FMI images. It is proven that GLCM can depict shale sedimentary structures efficiently and accurately, and four texture features (contrast, entropy, energy, and homogeneity) are sensitive to shale sedimentary structures. To accommodate the correlation between the four texture features, the random forest algorithm, which has been proven not to be affected by correlated input features, is selected for supervised lithofacies classification. To enhance the model's ability to differentiate between argillaceous limestone and calcareous mudstone, the carbonate content and clay content calculated from the ECS logs are involved in the input features. Moreover, grid search cross-validation (CV) is implemented to optimize the hyperparameters of the model. The optimized model achieves favorable performance on training data, validation data, and test data, with average accuracies of 0.84, 0.79, and 0.76, respectively. This study also discusses the application of the classification model in lithofacies and production prediction.

**Keywords:** shale lithofacies; sedimentary structure; FMI image; GLCM; texture features; random forest

## 1. Introduction

The Paleogene shale rich in organic matter in the Boxing Sag was deposited in a continental faulted lacustrine basin. In contrast with coarse clastic sedimentary rocks and marine shale, Paleogene shale in the Boxing Sag develops more lithofacies types [1–3]. In the past ten years, geologists have performed many studies on shale lithofacies identification and classification in the Boxing Sag [4]. Their results indicate that three main types of sedimentary structures (laminated structure, layered structure, and massive structure) are developed, and sedimentary structures of shale play a significant role in determining the reservoir properties [5–7]. Recent studies have proven that lithofacies with different sedimentary structures vary greatly in their physical and oil-bearing properties [8,9]. Lithofacies with laminated structures possess higher values of porosity, TOC (total organic carbon), and S1 (residual hydrocarbon). Hence, lithofacies with laminated structures are





identified as the most promising lithofacies types, followed by lithofacies with layered structures and lithofacies with massive structures. Therefore, the identification of shale sedimentary structures is an important part of lithofacies classification for Paleogene shale in the Boxing Sag.

Generally, shale lithofacies identification exploits petrophysical measurements, including wireline logs and laboratory core tests [10–12]. Due to limited core data in many instances, shale lithofacies are often classified from conventional wireline logs. In general, conventional wireline logging projects include caliper logging, gamma ray logging, self-potential logging, bulk density logging, neutron porosity logging, compressional waves sonic logging, and resistivity logging. Antariksa et al. predicted lithofacies, including shale, shaly sandstone, sandstone, and coal, from a well logging dataset in the Tarakan Basin, Indonesia [13]. Bhattacharya et al. classified five mudstone lithofacies, along with calcareous siltstone and limestone lithofacies from conventional well log suites (gamma ray, deep resistivity, bulk density, neutron porosity, and photoelectric factor log) [14]. Kim et al. defined four lithofacies in Eagle Ford shale (organic-matter-rich mudstone, organic-matter-lean calcareous marl, heterogeneous argillaceous wacke stone, and marl and massive marly chalk) and Lower Austin chalk using five wireline logs (gamma ray, bulk density, neutron porosity, deep resistivity, and compressional sonic logs) [15].

However, previous studies have mainly focused on the interpretation of shale mineral components or the classification of lithologies rather than the identification of shale sedimentary structures. This is mainly attributed to the scale mismatch between the shale sedimentary structures and the vertical resolutions of conventional wireline logs. In general, shale laminae are millimeter-scale, while the highest vertical resolution of conventional wireline logs is 0.5 m, which makes it impossible to identify sedimentary structures in conventional wireline logs.

Among current logging technologies, FMI logging has the highest vertical resolution, 5 mm, providing clear images of sedimentary structures, and it is an ideal tool for discriminating sedimentary facies and lithologies [16–18]. With the rapid development of image processing techniques, various texture analysis approaches have emerged, including the Tamura method, LBP (local binary pattern), GLCM (gray level co-occurrence matrix), wavelet transform, autocorrelation function, etc. Based on these methods, a growing number of studies related to feature extraction from FMI images have been performed. Zhang et al. extracted 69 texture features from FMI images using the autocorrelation function method to identify rock types in quartz sandstone reservoirs and achieved an accuracy of 78% [19]. Wang Man et al. successfully distinguished five types of volcanic rocks from FMI images by adopting GLCM [20]. Chai et al. employed LBP for feature extraction from FMI images to identify lithology in reef-bank reservoirs [21]. Luo et al. extracted four texture features using GLCM on FMI images to classify sedimentary microfacies in the gravel reservoir [22]. Yan et al. utilized the image-connected domain labeling algorithm to mark the solution-hole-connected domain from FMI images and obtain the heterogeneity information about solution hole in the carbonate reservoir [23]. Shafiabadi et al. detected fractures in FMI images using the Canny and Sobel algorithms as edge detection operators [24]. Wang Min et al. successfully transformed FMI images into continuous core gravel information after performing filtering with a median filter, image segmentation through the gray threshold segmentation algorithm, and gravel extraction through the Hoshen–Kopelman algorithm [25]. From Table 1, it can be concluded that at present, image feature extraction methods are mostly used to detect gravel, fracture, and classify lithologies from FMI images, with few studies on the division of sedimentary structures and even fewer applications in shale.

On the other hand, shale lithofacies possess relatively consistent petrophysical properties in contrast to conventional reservoirs, which causes some shale lithofacies interpreted from core data to not be distinguishable from wireline logs. To address this problem, machine learning algorithms have been increasingly introduced in shale lithofacies classification. With the ability to learn from large datasets, machine learning algorithms can grasp

data features and discover patterns in data [26–29]. The commonly used classification algorithms include support vector machine (SVM), K-nearest neighbors (KNN), naive Bayes (NB), decision tree (DT), artificial neural network (ANN), AdaBoost, and random forest (RF).

**Table 1.** Application of texture analysis methods on FMI images.

| Texture Analysis Methods | Reference | Year | Reservoir | Application |
|---|---|---|---|---|
| Autocorrelation function | [19] | 2014 | Quartz sandstone | Extracted 69 texture features to identify rock types |
| GLCM | [20] | 2009 | Volcanic rock | Distinguished five types of volcanic rocks |
| LBP | [21] | 2009 | Reef-bank | Identified sedimentary facies and lithology |
| GLCM | [22] | 2023 | Gravel | Classified four types of sedimentary microfacies |
| Image-connected domain labeling algorithm | [23] | 2016 | Carbonate | Extracted solution pore information |
| Canny and Sobel algorithm | [24] | 2021 | Fracture | Detected fractures |
| Median filter, gray threshold segmentation algorithm, Hoshen–Kopelman algorithm | [25] | 2019 | Gravel | Extracted core gravel features |

Bhattacharya et al. used an SVM to recognize the pattern of different shale lithofacies associated with basic petrophysical parameters from conventional well log suites [30]. Kim et al. trained a convolutional neural network (CNN) model to classify the lithofacies of Eagle Ford shale from five wireline logs [15]. Liu et al. employed an ANN approach to better understand the primary factors that control lacustrine shale lithofacies development [31]. Merembayev et al. compared machine learning algorithms, including KNN, DT, RF, eXtreme gradient boosting (XGBoost), and LightGBM, in lithofacies classification from various well log data from Kazakhstan and Norway. The random forest model had the best score among the considered algorithms [32]. Hoang et al. used the random forest algorithm to predict the lithofacies of the Balder field from well logs and seismic data and obtained favorable outcomes [33]. Antariksa et al. compared several machine learning algorithms in classifying lithofacies in the Tarakan Basin, Indonesia. Random forest and gradient boosting outperformed the other models in the experiment, with accuracies of 87.49% and 87.01%, respectively [13]. The abovementioned applications are summarized in Table 2.

**Table 2.** Performance of machine learning models in lithofacies classification.

| Machine Learning Algorithms | Reference | Year | Accuracy |
|---|---|---|---|
| SVM | [30] | 2016 | 87.3% |
| CNN | [15] | 2022 | 77.7% |
| ANN | [31] | 2020 | 88.6% |
| KNN, DT, XGBoost, LightGBM, RF | [32] | 2021 | RF (93.8%) LightGBM (90.1%) |
| RF | [33] | 2022 | 91.1% |
| DT, RF, Gradient Boosting, KNN, SVM, Logistic Regression | [13] | 2022 | RF (87.5%) Gradient Boosting (87.0%) |

In this work, we aim to introduce a convenient and effective approach for the classification of shale lithofacies with different sedimentary structures from FMI logs and ECS logs. This method combines the GLCM technique and a machine learning algorithm. First, the texture features that can quantitatively characterize the shale sedimentary structure are extracted from FMI images by adopting the GLCM technique. A sensitivity analysis of texture features is conducted to assure adaptability. Afterward, a dataset for training the classification model is constructed. This dataset consists of texture features as well as mineral contents from ECS logs. Finally, a hyperparameter-optimized random forest classifier is applied for automatic lithofacies classification.

## 2. Geological Setting

The Boxing Sag located in the southwest of the Dongying Depression is a graben basin controlled by faults. The whole area is characterized by the development of basin-dipping faults and reverse parallel faults (Figure 1a,b). The sag is filled with the Kongdian Formation, Shahejie Formation, and Dongying Formation of the Paleogene System; the Guantao Formation and Minghuazhen Formation of the Neogene System; and the Pingyuan Formation of the Quaternary System. The Shahejie Formation is the principal oil-producing zone, and it is subdivided into four members ($Es_4$, $Es_3$, $Es_2$, and $Es_1$, from bottom to top) (Figure 1d). The early deposition period of $Es_4$ to the late deposition period of $Es_3$ is the main active period of fault subsidence. During this period, the water depth of the lake basin increases, and there are multiple lake transgression and regression cycles, which form a series of lacustrine shale deposits. The total thickness of the lacustrine shale is more than 1000 m, and it is the main source rock of the Boxing Sag. The lithology is mainly mudstone, limestone, calcareous marl, and oil shale. There are 69 evaluation wells in the sag, two of which have core samples (Figure 1c). The previous analysis suggests that the target lower third and upper fourth members of the Shahejie Formation in the Boxing Sag exhibit great shale oil exploration potential [34].

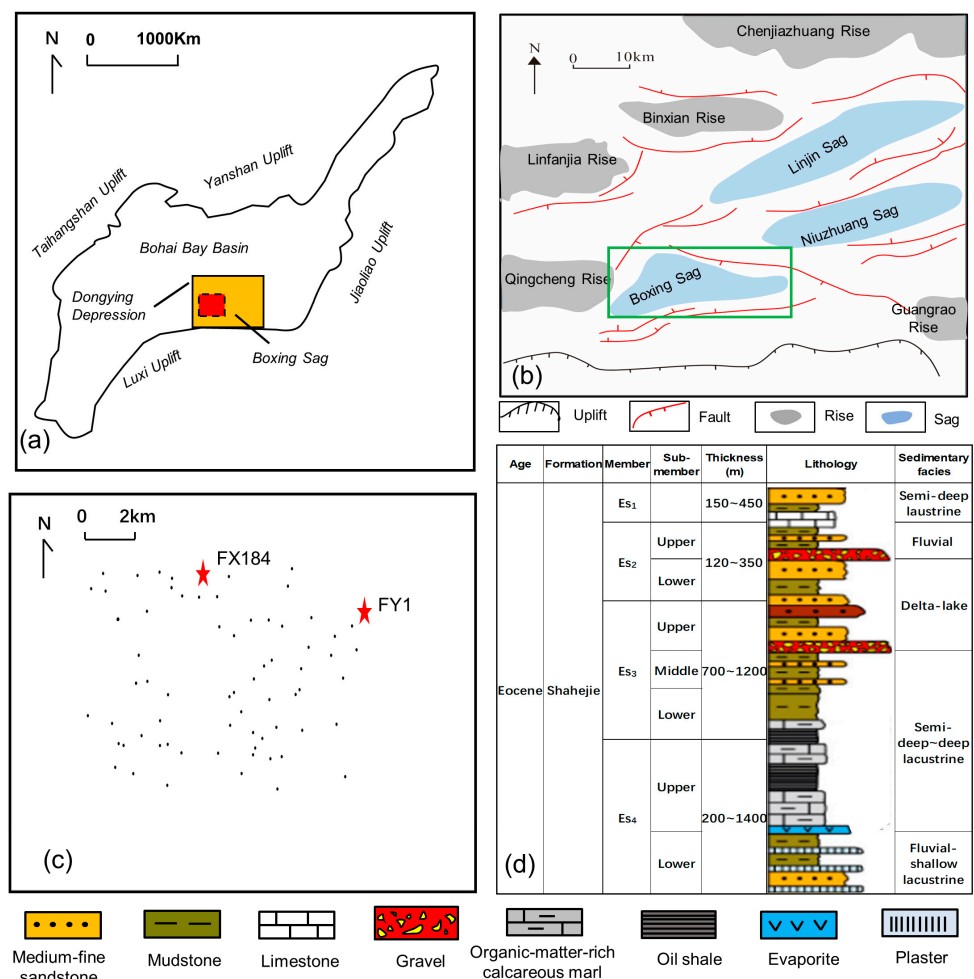

**Figure 1.** Geological maps: (**a**) Sketch map of the Bohai Bay Basin; (**b**) tectonic division of the Dongying Depression; (**c**) locations of the evaluation wells in the Boxing Sag; (**d**) section of the stratigraphic subdivision of the study area (adapted from [3]).

## 3. Data

### 3.1. Data Acquisition and Preprocessing

Two cored shale wells were utilized for this study: well FY1 and well FX184. The length of the cored interval is 191.7 m for well FY1 and 239.7 m for well FX184. A total of 359 core samples from well FY1 and 201 core samples from well FX184 were tested. X-ray diffraction (XRD) data, thin sections, and scanning electron microscopy (SEM) images were used for lithofacies analysis.

Conventional wireline logs were run in both wells, including spontaneous potential, gamma ray, caliper, compressional sonic log, neutron porosity, bulk density, deep resistivity, and shallow resistivity logs, as shown in the second column to the fourth column in Figure 2. The fifth column demonstrates the core-interpreted sedimentary structures based on the XRD data. The cross plots in the sixth column show that there are no distribution patterns for the three types of sedimentary structures in the conventional cross plots, indicating that it is difficult to distinguish shale sedimentary structures from conventional wireline logs.

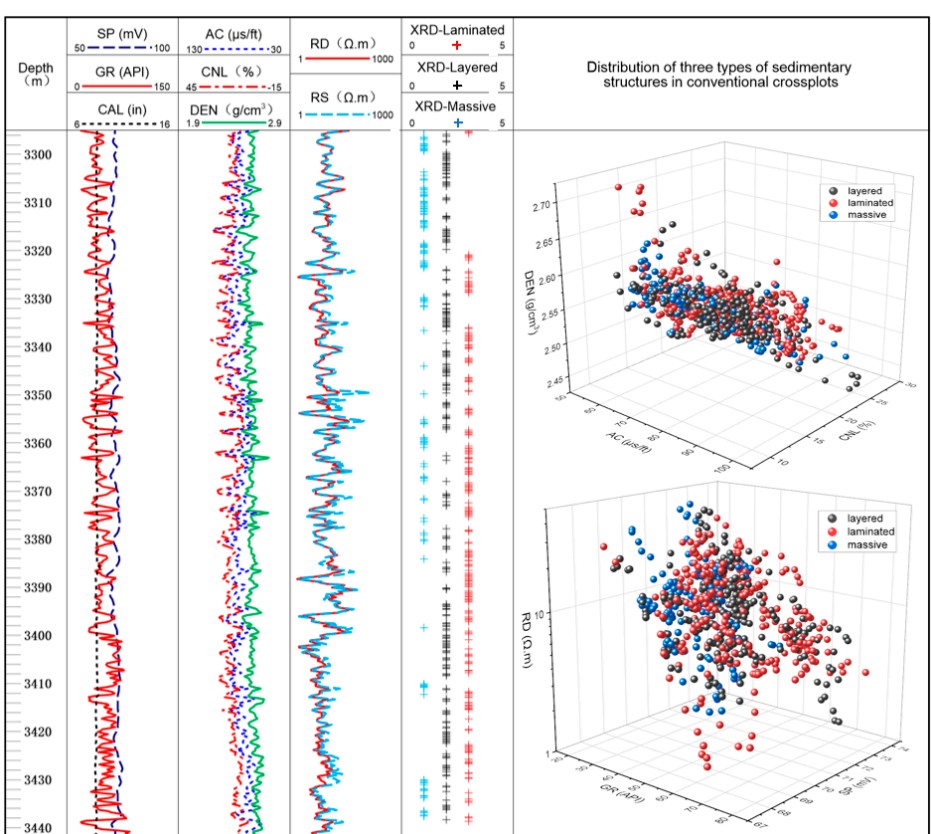

**Figure 2.** Conventional wireline logs, sedimentary structure from XRD and cross plots of well FY1.

In addition to conventional wireline logs, the two wells both have FMI logs and ECS logs. The measured intervals of the two wells are 3250 m~3413 m for well FY1 and 3400 m~3614 m for well FX184. Mineral contents from the ECS logs are described in Figure 3 and Table 3.

Data preprocessing was implemented before lithofacies identification. First, depth calibration was conducted for the FMI logs and ECS logs. Then, the FMI static images were converted to grayscale images with 230 pixels in length and width, and the gray level was set to 16. A total of 652 FMI grayscale images were obtained from well FY1, and 842 FMI grayscale images were obtained from well FX184.

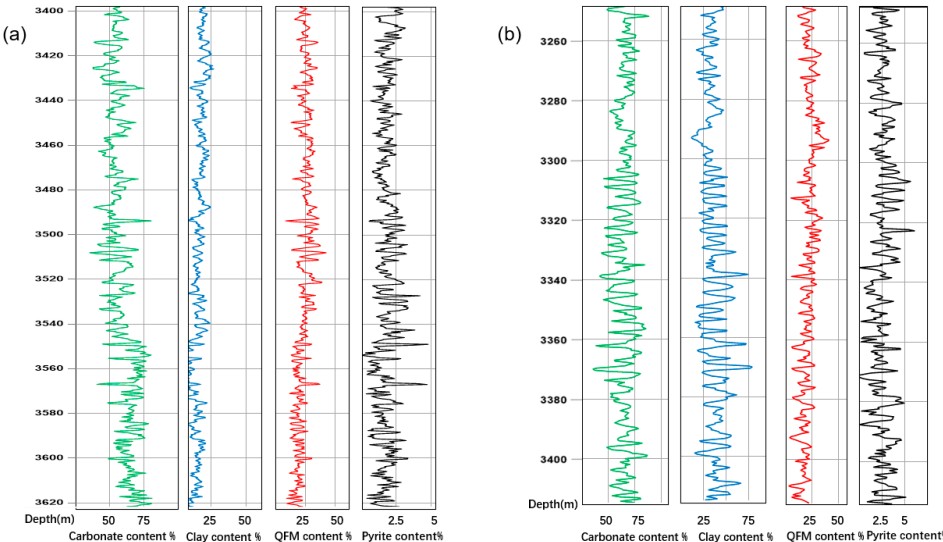

**Figure 3.** Mineral contents from ECS logs of (**a**) well FX184 and (**b**) well FY1.

**Table 3.** Mineral content distribution from ECS logs.

| Mineral Content % | FY1 | | | FX184 | | |
|---|---|---|---|---|---|---|
| | **Min** | **Max** | **Mean** | **Min** | **Max** | **Mean** |
| Carbonate | 10.2 | 63.8 | 40.1 | 35.9 | 80.3 | 57.9 |
| Clay | 12.2 | 79.0 | 35.9 | 4.4 | 26.8 | 16.3 |
| QFM [1] | 3.2 | 41.3 | 21.5 | 8.0 | 42.4 | 24.0 |
| Pyrite | 0 | 6.6 | 2.5 | 0 | 5.2 | 1.8 |

[1] QFM = quartz, feldspar, and mica.

### 3.2. Lithofacies Classification

According to the sedimentary structure, the Paleogene shale in the Boxing Sag is divided into three categories: laminated, layered, and massive. The shale is close in mineral composition. The main minerals are clay minerals and carbonate. Therefore, the lithology is defined by clay mineral content and carbonate content by the cut-offs of 50% and 25%. When the clay content is greater than 50% and the carbonate content is greater than 25% and less than 50%, shale is defined as calcareous mudstone. When the carbonate content is greater than 50% and the clay mineral content is greater than 25% and less than 50%, it is defined as argillaceous limestone. When the carbonate content is greater than 50% and the clay mineral content is less than 25%, it is defined as marly limestone. In this way, the shale in the study is divided into three kinds of lithology: argillaceous limestone, calcareous mudstone, and marly limestone. The definitions of five lithofacies can be concluded in the following Table 4.

**Table 4.** Definition of the five lithofacies.

| Lithofacies Types | Sedimentary Structure | Clay Mineral Content | Carbonate Content |
|---|---|---|---|
| Lithofacies 1 | Laminated | 25~50% | ≥50% |
| Lithofacies 2 | Laminated | ≥50% | 25~50% |
| Lithofacies 3 | Layered | 25~50% | ≥50% |
| Lithofacies 4 | Layered | ≥50% | 25~50% |
| Lithofacies 5 | Massive | ≤25 | ≥50% |

#### 3.2.1. Lithofacies 1: Laminated Argillaceous Limestone

This lithofacies is formed in saline standing water with stable conditions for the stratification of water bodies and the preservation of organic matter. The core is dark

gray and consistent interbedded organic-rich dark argillaceous laminae and bright calcite laminae appear. The calcite laminae are relatively thicker than the argillaceous laminae (Figure 4a,d). The FMI image shows obvious alternate bright and dark bands (Figure 4b). The mineral composition consists of carbonate (calcite and dolomite), clay minerals, and small amounts of quartz and pyrite (Figure 4e). The clay mineral content and carbonate content from ECS logs are shown in Figure 4c. The carbonate content is between 41% and 72%, the clay mineral content is between 28% and 52%, and the quartz content is between 4% and 21%. The pore types include clay mineral intragranular pores, dissolution pores, intragranular pores, intergranular quartz feldspar pores, organic matter pores, and calcite intergranular pores. The pore diameter is usually less than 3 μm. The organic matter content is high, and the TOC is between 3% and 4%.

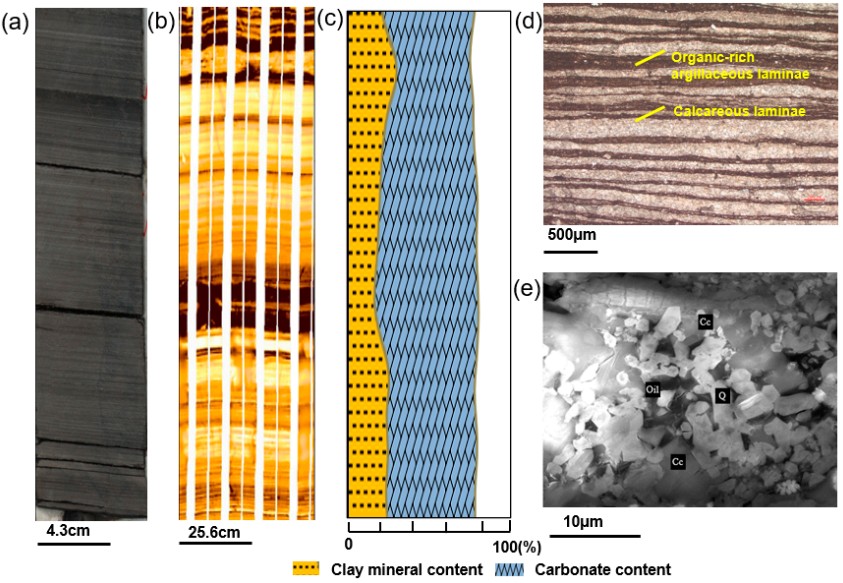

**Figure 4.** Lithofacies 1 (laminated argillaceous limestone): (**a**) core; (**b**) FMI image; (**c**) mineral contents from ECS logs; (**d**) thin section: interbedded organic-rich dark argillaceous laminae and bright calcite laminae; (**e**) SEM image: calcite surface filled with quartz particles and crude oil. Cc = calcite; Q = quartz.

### 3.2.2. Lithofacies 2: Laminated Calcareous Mudstone

This core is dark black and laminated, but the boundaries between the layers are not as evident as those in Lithofacies 1 (Figure 5a). The FMI image is characterized by frequent alternate light and dark bands (Figure 5b). As shown in thin section (Figure 5d), laminae are clearly identified and are mainly composed of organic-rich argillaceous laminae and calcite-rich mixed laminae. The clay mineral content ranges from 39% to 75%, the carbonate content lies between 23% and 51%, and the quartz content is between 2% and 18% compared with those of Lithofacies 1 (Figure 5c,e).

### 3.2.3. Lithofacies 3: Layered Argillaceous Limestone

This lithofacies is mainly formed in a reducing environment with humid climate and deep water. Due to the weak seasonal climate control on deposition, the shale is dominated by a layered structure. The core is light gray with indistinct layers (Figure 6a). The dark bands are intermittently exhibited in the FMI image (Figure 6b). As displayed in thin section (Figure 6d), the layers can be observed distinctly, and the micritic calcite layers are interbedded with mineral-mixed layers. The pore types are mainly clay mineral intergranular pores and calcite intergranular pores, as well as calcite dissolution pores and a small amount of quartz feldspar intergranular pores. The carbonate content is between 42% and 78%, the clay mineral content is between 21% and 47%, and the quartz content is

between 7% and 19%. (Figure 6c,e). The abundance of organic matter is high, and the TOC is distributed between 2% and 4%.

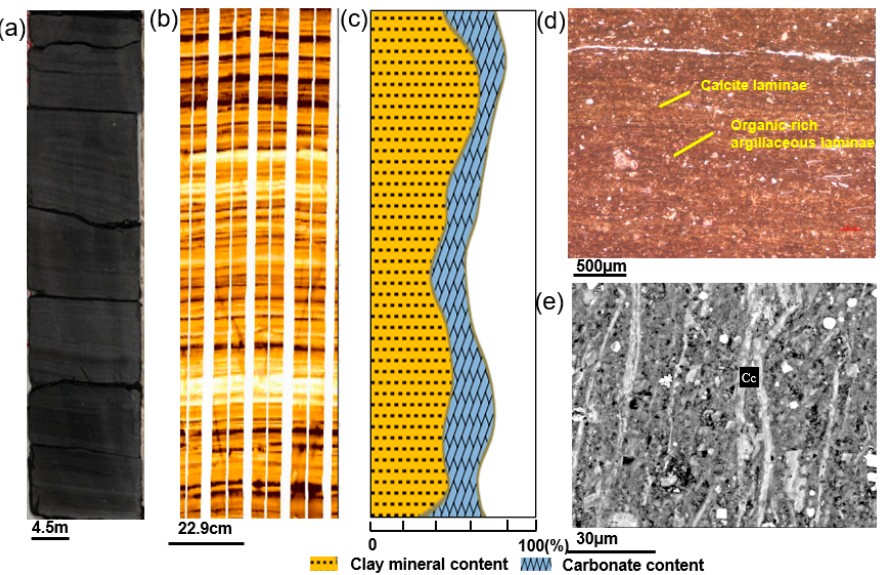

**Figure 5.** Lithofacies 2 (laminated calcareous mudstone): (**a**) core; (**b**) FMI image; (**c**) mineral contents from ECS logs; (**d**) thin section: organic-rich clay laminae and calcite-rich mixed laminae; (**e**) SEM image: banded calcite. Cc = calcite.

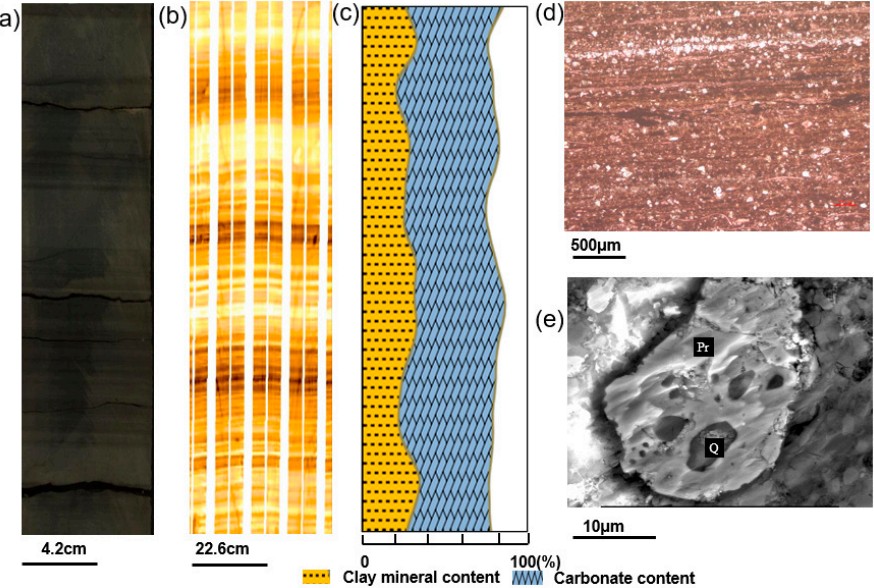

**Figure 6.** Lithofacies 3 (layered argillaceous limestone): (**a**) core; (**b**) FMI image; (**c**) mineral contents from ECS logs; (**d**) thin section: layered calcite; (**e**) SEM image: pores filled by pyrite, quartz, and crude oil. pr = pyrite; Q = quartz.

### 3.2.4. Lithofacies 4: Layered Calcareous Mudstone

The core is dark gray with faintly visible layers, and the argillaceous layer is relatively thick (Figure 7a). The bright bands intermittently appear in the FMI image (Figure 7b). As observed under a microscope, clay, feldspar, quartz, and calcite are mixed, and calcite is locally rich with a lenticular or layered distribution (Figure 7d). Feldspar and quartz are mostly dispersedly distributed. The clay mineral content increases, generally between 48% and 82%, with an average of 68%, compared with that of Lithofacies 3. The carbonate

content ranges from 27% to 49%, and the quartz content is between 5% and 11% (Figure 7c,e). The abundance of organic matter is close to that of Lithofacies 3.

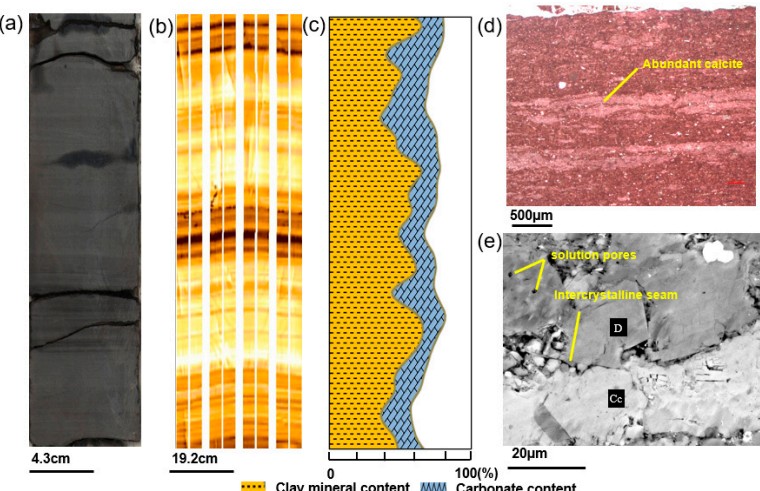

**Figure 7.** Lithofacies 4 (layered calcareous mudstone): (**a**) core; (**b**) FMI image; (**c**) mineral contents from ECS logs; (**d**) thin section: calcite minerals are partly mixed with other minerals and partly layered abundant; (**e**) SEM image: dissolution pores and intercrystal seam are identified. Cc = calcite; D = dolomite.

### 3.2.5. Lithofacies 5: Massive Marly Limestone

The lithofacies is mainly developed in a dry climate and a saline water environment with fewer material sources. The core is light gray without laminae, and abundant ostracod fragments and scattered quartz silt particles can be observed (Figure 8a,d). The FMI image exhibits a bright yellow block (Figure 8b), and the quartz feldspar content is between 4% and 15%, with traces of pyrite (Figure 8c,e). The bedding fractures are not developed; generally only structural fractures are developed. The pore types are mainly intergranular pores with poor connectivity. The content of organic matter is low, and the TOC is mainly between 1% and 2%.

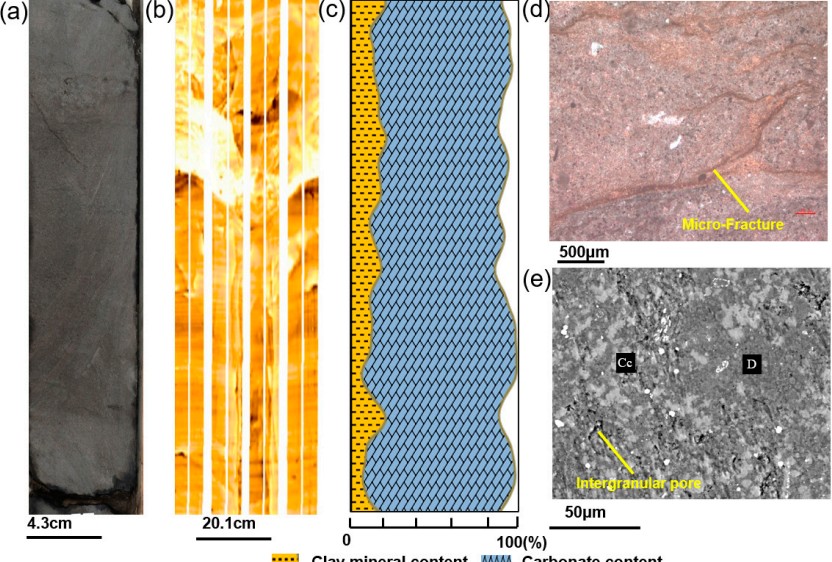

**Figure 8.** Lithofacies 5 (massive marly limestone): (**a**) core; (**b**) FMI image; (**c**) mineral contents from ECS logs; (**d**) thin section: microfracture is identified; (**e**) SEM image: main minerals are calcite and dolomite. Cc = calcite; D = dolomite.

## 4. Methods

GLCM measures image texture features based on the spatial relationship of pixels. The laminae of shale are mostly parallel, exhibiting parallel textures with various gray scales in grayscale FMI images. The different sedimentary structures of shale demonstrate different spatial correlation characteristics of gray scales in FMI images, which provide the basis for the application of GLCM.

### 4.1. Gray Level Co-Occurrence Matrix (GLCM)

The gray level co-occurrence matrix of an image can be obtained by first counting the frequency of an element P (*i*, *j*, *d*, *θ*) of the image and then transforming the frequency into probabilities. These steps are performed by dividing by the total frequency of all elements, where *i* is the gray level of a pixel at location (*x*, *y*) and *j* is the gray level of a neighboring pixel at location (*x* + *dx*, *y* + *dy*). This relationship of the two pixels is defined by two parameters: offset, *d*, and orientation, *θ* (Figure 9). GLCMs with different (*d*, *θ*) combinations capture different information related to the textural appearance of an image [35].

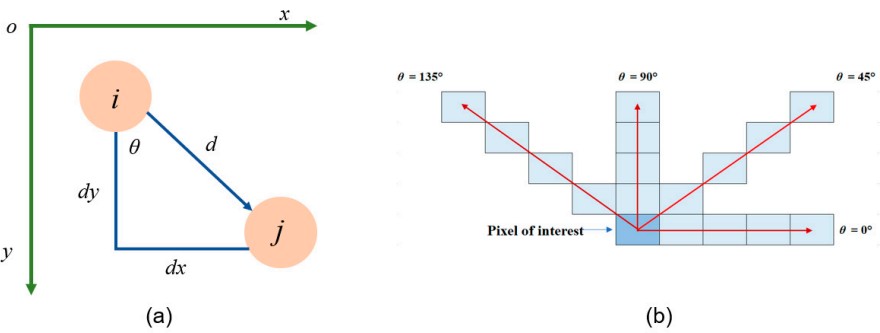

**Figure 9.** (**a**) Pixel–pair relation and (**b**) four types of orientation of the GLCM.

The size of the GLCM depends on the gray level value of an image. Suppose *L* is the gray level value of an image; then, its GLCM is an $L \times L$ dimensional matrix. In general, considering the amount of matrix computation and the quality of the image texture, the grayscale *L*-value is usually chosen to be 8 or 16.

After computing the GLCM, different features can be obtained. Haralick proposed the extraction of 14 statistical features from a GLCM [36]. It has been found that 5 features of the 14 statistical features, including energy, contrast, entropy, homogeneity, and correlation, are not only easy to calculate but can yield higher classification accuracy [37–39]. The five properties are explained below along with the mathematical equations used.

### 4.1.1. Energy

$$Energy = \sum_{i=0}^{L-1} \sum_{j=0}^{L-1} P(i,j)^2 \tag{1}$$

The energy measures the homogeneity of an image. The more uniform the image, the greater the value of energy.

### 4.1.2. Contrast

$$Contrast = \sum_{i=0}^{L-1} \sum_{j=0}^{L-1} (i-j)^2 P(i,j) \tag{2}$$

The contrast reflects the intensity of the difference between the neighboring pixels in the co-occurrence matrix. It varies between the largest and smallest values in a continuous group of pixels. The value of contrast for a constant image is 0.

4.1.3. Homogeneity

$$Homogeneity = \sum_{i=0}^{L-1}\sum_{j=0}^{L-1} \frac{P(i,j)}{1+(i-j)^2} \tag{3}$$

The homogeneity measures the similarity of pixel values. The range of homogeneity varies between 0 and 1. It has the highest value when all the pixel values in an image are alike.

4.1.4. Correlation

$$Correlation = \sum_{i=0}^{L-1}\sum_{j=0}^{L-1} \frac{(ij)P(i,j) - u_i u_j}{S_i S_j} \tag{4}$$

where

$$u_i = \sum_{i=0}^{L-1} i \sum_{j=0}^{L-1} P(i,j) \tag{5}$$

$$u_j = \sum_{i=0}^{L-1} j \sum_{j=0}^{L-1} P(i,j) \tag{6}$$

$$S_i^2 = \sum_{i=0}^{L-1} (i-u_i)^2 \sum_{j=0}^{L-1} P(i,j) \tag{7}$$

$$S_j^2 = \sum_{i=0}^{L-1} (j-u_j)^2 \sum_{j=0}^{L-1} P(i,j) \tag{8}$$

The correlation describes how closely the neighboring pixels are connected. The range of correlation is between −1 and 1. A value of −1 specifies a perfectly negative correlation, while a value of 1 means a perfectly positive correlation.

4.1.5. Entropy

$$Entropy = -\sum_{i=0}^{L-1}\sum_{j=0}^{L-1} P(i,j) \cdot log P(i,j) \tag{9}$$

The entropy measures the overall information about an image. The entropy value is low for an irregular co-occurrence matrix.

The main processing steps of GLCM for FMI images are described in Figure 10. First, a raw FMI static image is converted into a grayscale image, and then a gray matrix of the image is obtained. Afterward, the gray level co-occurrence matrix of the image is calculated. Finally, five texture features are extracted from the GLCM.

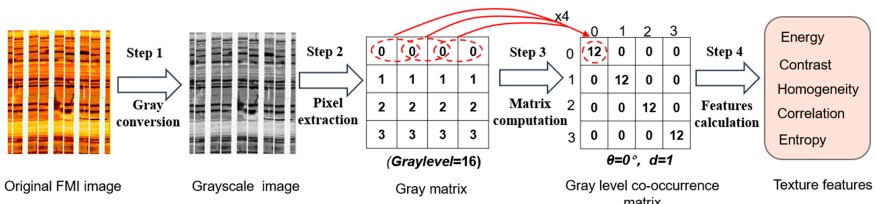

**Figure 10.** Diagram of the main steps of the GLCM for FMI images.

### 4.2. Random Forest (RF)

The random forest algorithm is an ensemble of decision trees that can be applied to classification or regression tasks. The random forest algorithm adopts the bootstrap aggregating technique (known as "bagging"). Each decision tree in the random forest model is trained by bootstrap samples of input data, which reduces the correlation between decision trees. The final classification result of a random forest model is decided by the majority vote from all decision trees [40–43].

The steps for building a classification tree in the random forest model are as follows:

(1) After preprocessing the training data, $n$ ($n < N$) samples are randomly selected from input dataset $N$. Each decision tree is trained on a different subset of the training data.

(2) If the number of input features is $M$, a constant $m$ ($m << M$) is assigned, and $m$ variables are randomly selected from $M$ features. When a node splits, the feature with the highest purity is selected from the m features after calculating the Gini index for each feature. The lower the Gini index, the higher the purity of the feature. The Gini index of feature $K$ in dataset $D$ is calculated as follows [44]:

$$Gini\_index(D, k) = \sum_{v=1}^{V} \frac{|D^v|}{|D|} Gini(D^v) \tag{10}$$

$$Gini(D^v) = \sum_{i=1}^{n} \sum_{i' \neq i} p_i p'_i = 1 - \sum_{i=1}^{n} p_i^2 \tag{11}$$

where $V$ is the number of subsets based on feature $K$; $D^v$ is a subset of dataset $D$ on feature $k$; $|D^v|$ and $|D|$ are the total numbers of samples in subset $D^v$ and in dataset $D$, respectively; Gini ($D^v$) is the Gini value of subset $D^v$; $n$ is the number of types in dataset $D^v$, and $p$ is the proportion that type $i$ occurs in dataset $D$.

(3) The decision tree is fully grown and not pruned. Node splits are typically continued until nodes are pure (one class).

### 4.3. Proposed Lithofacies Classification Model

After presenting the principles of GLCM and RF, an integrated approach for lithofacies classification is proposed. This approach utilizes GLCM to extract texture features from FMI images and then inputs both the texture features and mineral content calculated from the ECS log to an RF classifier. After the identification of each decision tree in the RF classifier, the result of lithofacies classification is received (Figure 11).

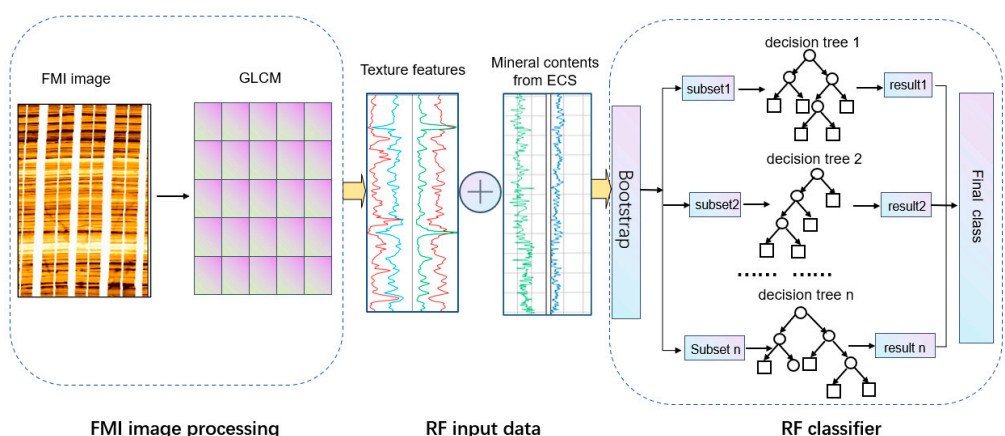

**Figure 11.** Workflow of the GLCM–RF model for lithofacies classification.

### 4.4. Criteria for Verifying the Model Performance

The prediction performance of the model is evaluated using four statistical quality indicators, including precision, recall, F1-score, and accuracy. These indicator values are in

the range [0, 1]. The higher the value, the better the model performs [45]. The indicators are defined as follows:

$$Precision = \frac{TP}{TP + FP} \tag{12}$$

$$Recall = \frac{TP}{TP + FN} \tag{13}$$

$$\frac{1}{F1} = \frac{1}{2}\left(\frac{TP + FP}{TP} + \frac{TP + FN}{TP}\right) \tag{14}$$

$$Accuracy = \frac{TP + TN}{TP + TN + FP + FN} \tag{15}$$

where TP represents the correct classifications in the positive class, TN represents the correct classifications in the negative class, FP represents the incorrect classifications in the positive class, and FN represents the incorrect classifications in the negative class.

*4.5. Hyperparameter Tuning*

The previous studies show that the complexity of the dataset directly affects the performance of the machine learning models. As for the application of lithofacies classification from well logs, the complexity of the dataset is mainly reflected in the number of features [46]. The performance of machine learning models does not always improve with the increase in the number of features. The number of features required for a machine learning model is an open question. In general, simple machine learning models with fewer features are easier to understand and interpret, and overfitting can be avoided. In this study, considering the negative impact of a complex dataset on the accuracy of the model, six input features were selected for lithofacies classification.

The performance of any machine learning model also highly depends on the selection of the model hyperparameters [47]. The hyperparameters of the random forest algorithm mainly consist of the number of decision trees (n_estimators), maximum features (max_features), maximum depth of a tree (max_depth), minimum samples for a node to split (min_sample_split), and minimum samples for leaf nodes (min_samples_leaf). Among them, n_estimators can significantly impact the overall accuracy of the model. If the value of n_estimators is too low, the model may suffer from underfitting, while if the value of n_estimators is too high, the model performance cannot be significantly improved.

The best combination of hyperparameters needs to be tuned during a trial-and-error process. Considering that the dimensionality of the training dataset is not high in this study, a grid search CV is employed to ascertain the most promising hyperparameter combination [48]. The grid search approach covers the entire search space and tests for every possible combination of hyperparameters. In this study, a broad search with a larger step size of the hyperparameter space is first performed, and then a second, more refined search is conducted within a limited search space. For cross-validation, 10-fold cross-validation is selected. The whole dataset is divided into 10 folds. The 10th fold is used to test the model, and the remaining 9 folds are used for training.

The search space and step size for the considered hyperparameters are displayed in Table 5. Since the minimum step size is adopted, the best values for max_features, max_depth, min_samples_split, and min_samples_leaf can be obtained after the first search. For n_estimators, its relationship with the model accuracy during the first broad search is shown in Figure 12a. It is obvious that the model accuracy does not rise consistently with an increase in n_estimators; instead, it begins to decrease when n_estimators is greater than a certain value. When n_estimators is 71, the accuracy reaches the highest score of 0.941. Considering a step size of 10, the optimal value for n_estimators should be between 60 and 80. Then, a second refined search for optimal n_estimators is performed. The search range is narrowed down to [60, 80], and the search step size is set to 1. Figure 12b shows that

when n_estimators is 74, the model accuracy is the highest. Thus, after the fine search, the optimum value for n_estimators can be designated as 74.

**Table 5.** The first hyperparameters search result.

| Hyperparameters | Symbol | Search Space | Step Size | Optimal Value |
|---|---|---|---|---|
| Number of decision trees to fit | n_estimators | [1, 100] | 10 | 74 |
| Maximum features | max_features | [1, 6] | 1 | 3 |
| Maximum depth of a tree | max_depth | [1, 10] | 1 | 5 |
| Maximum samples for a node to split | min_sample_split | [1, 5] | 1 | 1 |
| Minimum samples for leaf nodes | min_samples_leaf | [1, 5] | 1 | 2 |

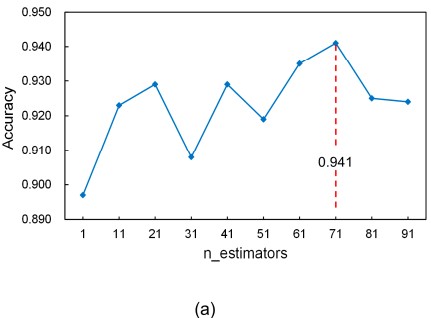

(a)

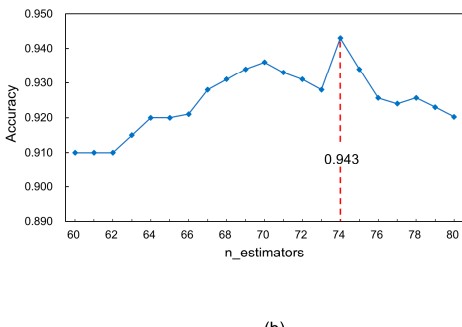

(b)

**Figure 12.** Relationship between the model accuracy and n estimators in a (**a**) broad search and (**b**) fine search.

*4.6. Data Split Sensitivity*

The impact of different data split ratios is tested. The ratio of training data to validation data is designed from 50:50 to 90:10, with a 10% increase each time. As seen from Table 6, at the ratio of 50:50, the model achieves the highest accuracy score on the training set, but on the validation set and the test set, the model obtains the lowest accuracy scores, indicating the occurrence of an overfitting problem. As the proportion of the training set increases, the model accuracy on the training set gradually decreases, and the model accuracies on the validation set and the test set gradually increase. When the ratio is 80:20, the accuracies on the validation set and the test set are the highest values. Therefore, the training data ratio of 80:20 can be regarded as the optimal split ratio for the prediction model.

**Table 6.** Data split sensitivity.

| Data Split | Accuracy | | |
|---|---|---|---|
| | Train | Validation | Test |
| 50:50 | 0.94 | 0.70 | 0.68 |
| 60:40 | 0.93 | 0.70 | 0.67 |
| 70:30 | 0.91 | 0.71 | 0.68 |
| 80:20 | 0.89 | 0.74 | 0.73 |
| 90:10 | 0.88 | 0.70 | 0.69 |

*4.7. GLCM Texture Feature Sensitivity*

To assure the adaptability of extracted texture features from FMI images to the identification of shale sedimentary structures, a sensitivity analysis of five texture features is carried out.

A total of 50 typical FMI images, 10 FMI images for each lithofacies, are chosen from well FX184. The parameters of the GLCM are set as follows: L = 16; d = 1; $\theta = 0°, 45°, 90°$, and $135°$. Since four types of orientations are selected, four co-occurrence matrices are generated for each FMI image, and texture features of each FMI image are generated by averaging texture features from four kinds of orientations.

The visualized gray level co-occurrence matrices of the five lithofacies are exhibited in Figure 13. The gray level co-occurrence matrices demonstrate different numerical distribution characteristics for different types of lithofacies. For laminated lithofacies (Lithofacies 1 and Lithofacies 2), the values on the diagonal of the gray level co-occurrence matrices are lower compared with the values of layered lithofacies (Lithofacies 3 and Lithofacies 4), whereas the values on both sides of the diagonal are higher. The massive lithofacies (Lithofacies 5) has the highest value on the diagonal of the gray level co-occurrence matrix.

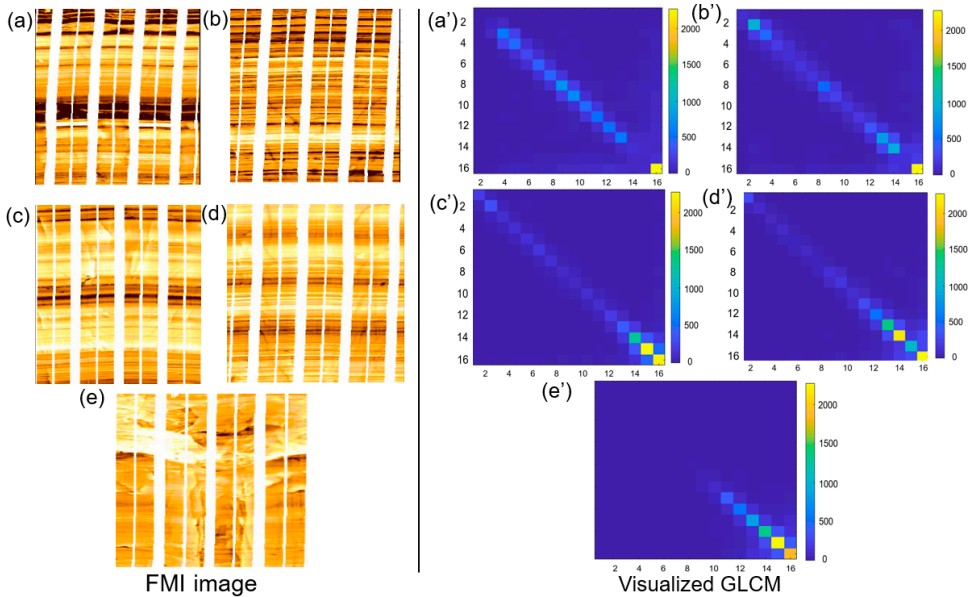

**Figure 13.** FMI images of (**a**) Lithofacies 1, (**b**) Lithofacies 2, (**c**) Lithofacies 3, (**d**) Lithofacies 4, and (**e**) Lithofacies 5 and visualized GLCMs of (**a′**) Lithofacies 1, (**b′**) Lithofacies 2, (**c′**) Lithofacies 3, (**d′**) Lithofacies 4, and (**e′**) Lithofacies 5.

In Table 7, the distribution ranges of five texture features in the five lithofacies can be observed. According to the statistics, for the laminated lithofacies (Lithofacies 1 and Lithofacies 2), contrast, correlation, and entropy are higher than in the other lithofacies, and the distribution ranges are 0.47~0.85, 0.54~0.90, and 0.67~0.95 respectively, while energy and homogeneity are lower than in the other lithofacies; their distribution ranges are 0.01~0.23 and 0.06–0.25 respectively. For massive lithofacies, the characteristics of the texture features are opposite to those of laminated lithofacies. Contrast, correlation, and entropy are lower than those of the other lithofacies, and the distribution ranges are 0.02~0.08, 0.08~0.44, and 0.03~0.28 respectively, while energy and homogeneity are higher than in the other lithofacies, with distribution ranges of 0.57~0.65 and 0.68~0.90 respectively. For the layered lithofacies (Lithofacies 3 and Lithofacies 4), the distribution ranges of texture features are between those of the laminated lithofacies and those of the massive lithofacies.

**Table 7.** Distribution ranges of texture features.

| Lithofacies Type | Contrast | Correlation | Energy | Homogeneity | Entropy |
| --- | --- | --- | --- | --- | --- |
| Lithofacies 1 | 0.75~0.85 | 0.58~0.90 | 0.01~0.12 | 0.06~0.19 | 0.90~0.95 |
| Lithofacies 2 | 0.32~0.40 | 0.54~0.85 | 0.15~0.30 | 0.34~0.43 | 0.67~0.73 |
| Lithofacies 3 | 0.47~0.70 | 0.18~0.70 | 0.07~0.23 | 0.20~0.25 | 0.72~0.87 |
| Lithofacies 4 | 0.17~0.27 | 0.36~0.93 | 0.52~0.80 | 0.46~0.60 | 0.55~0.58 |
| Lithofacies 5 | 0.02~0.08 | 0.08~0.44 | 0.57~0.65 | 0.68~0.90 | 0.03~0.28 |

Figure 14a shows the distribution map of five texture features in five types of sedimentary structures, indicating that contrast, entropy, energy, and homogeneity achieve better distinction among the five lithofacies than correlation, with a clear distribution range for each lithofacies. The cross plots referring to the correlation show no boundaries between different lithofacies, and data points representing different lithofacies are mixed together. Figure 14b shows that correlation is much more discrete than the other four features, and the overlap zone between the lithologies is larger, especially for the laminated facies and layered facies, which means that correlation is incapable of conducting lithofacies classification.

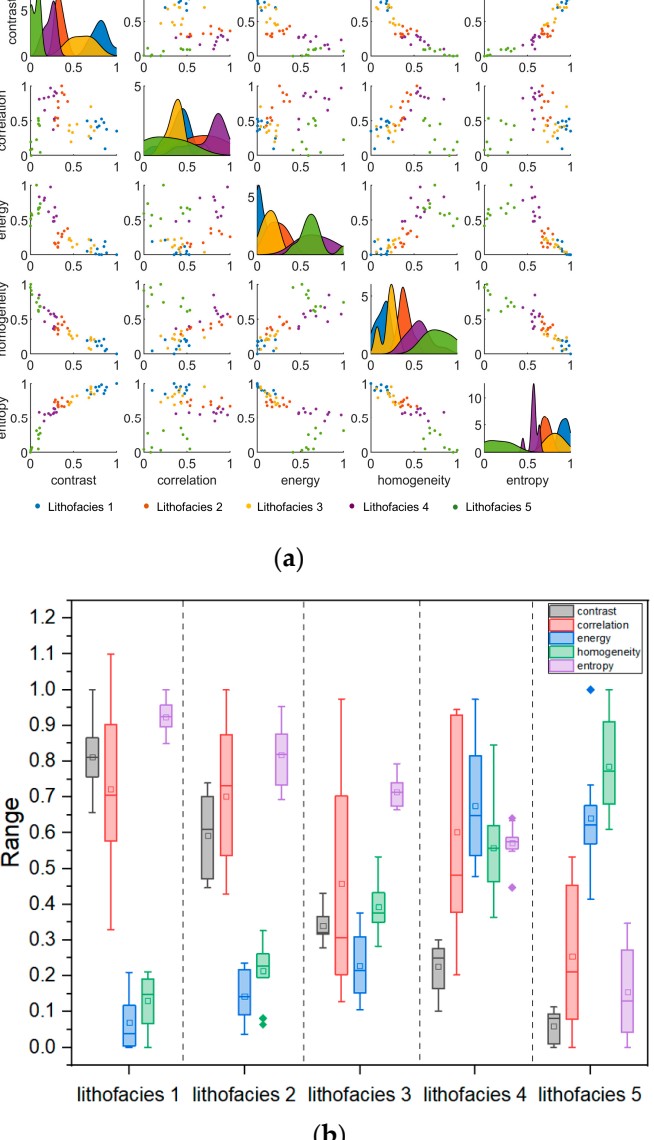

**Figure 14.** Visualized data distribution of five texture features. (**a**) Matrix diagram of texture features. (**b**) Box plot of texture features.

For further illustration, the GLCM texture feature curves calculated from the FMI images of well FX184 at the interval of 3491.8~3500.5 m are demonstrated in Figure 15. Four subintervals are selected for comparison. These subintervals are separately dominated by different types of sedimentary structures.

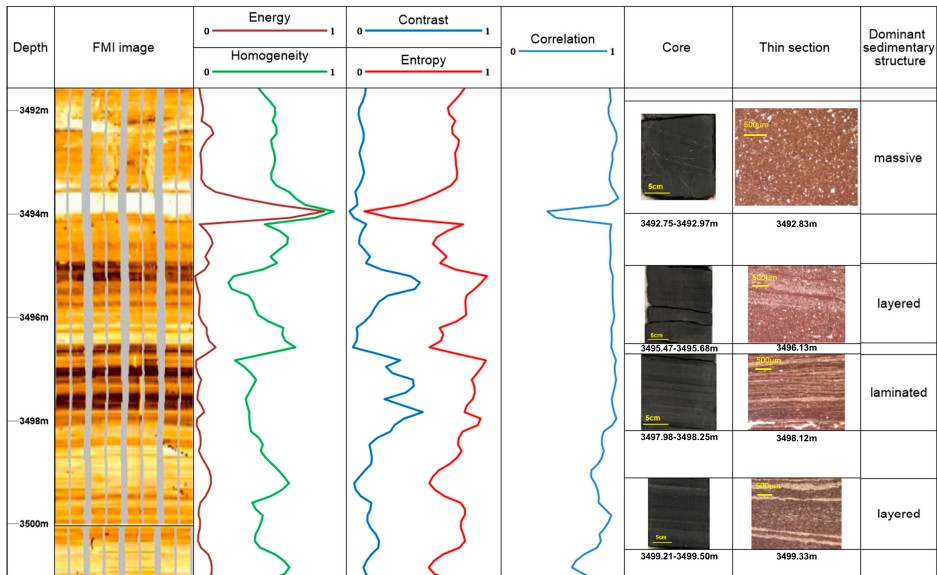

**Figure 15.** GLCM texture features of well FX184.

As shown in Figure 15, when the sedimentary structure changes from massive to laminated, contrast and entropy display an increasing trend, while homogeneity demonstrates the opposite trend. In contrast to other texture features, energy is more sensitive to the white band on the FMI images. Correlation shows the weakest correlation with the change in sedimentary structures, and the characteristic is consistent with it in cross plots from Figure 14.

### 4.8. Feature Correlation Analysis

The relationship between five texture features from the GLCM is investigated by calculating the Pearson correlation coefficient. The Pearson correlation coefficient reflects the degree of linear correlation between variables, and the range of the Pearson correlation coefficient varies between −1 and 1. The greater the absolute value, the stronger the correlation between the variables. Figure 16 shows that the values of the Pearson correlation coefficient between contrast, entropy, energy, and homogeneity are very high, all larger than 0.7, implying strong linear relationships within the four texture features.

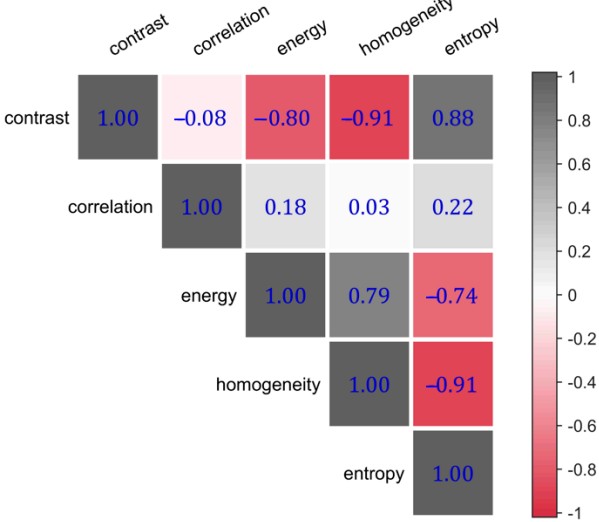

**Figure 16.** Correlation matrix of five texture features.

## 5. Results

The input variable set consists of four texture features (contrast, entropy, homogeneity, and energy) extracted from FMI images and two mineral contents (clay and carbonate) calculated from ECS logs. All variables are normalized to the range of [0, 1]. Samples from well FX184 are used for training and validating the model, and samples from well FY1 are used for a blind test. There are 635 samples from well FX184. The samples are divided into a training set and a validation set at a ratio of 80:20. There are 727 samples from well FY1, and these samples are used for testing the model. The numbers of samples for the five lithofacies are shown in Table 8.

**Table 8.** Numbers of samples.

| Well | Lithofacies 1 | Lithofacies 2 | Lithofacies 3 | Lithofacies 4 | Lithofacies 5 | Total |
|------|---------------|---------------|---------------|---------------|---------------|-------|
| FX184 | 114 | 172 | 176 | 129 | 44 | 635 |
| FY1 | 72 | 233 | 267 | 104 | 51 | 727 |

### 5.1. Comparison with Other Classifiers

The classification performances of classifiers for correlated input variables are examined between the RF model and several other classifiers (KNN, NBC, SVM, and DT). The models are trained on the initial dataset (Table 4) with default parameters. As demonstrated in Table 9, the results indicate that the SVM model achieves the lowest accuracy value and F1-score, followed by the NBC and KNN models. The DT model achieves better outcomes than those of the KNN model on both validation data and test data, although it suffers from an overfitting problem on the training set. The RF model outperforms all other classifiers, with the highest accuracy on distinct datasets, demonstrating its advantage in prediction with correlated input variables.

**Table 9.** Performance comparison of different classifiers.

| Model | Training Data | | Validation Data | | Test Data | |
|-------|----------|----------|----------|----------|----------|----------|
| | Accuracy | F1-Score | Accuracy | F1-Score | Accuracy | F1-Score |
| KNN | 0.59 | 0.71 | 0.51 | 0.60 | 0.42 | 0.55 |
| NBC | 0.42 | 0.41 | 0.42 | 0.44 | 0.31 | 0.37 |
| SVM | 0.22 | 0.38 | 0.26 | 0.44 | 0.28 | 0.51 |
| DT | 1.00 | 1.00 | 0.54 | 0.58 | 0.50 | 0.59 |
| RF | 0.81 | 0.84 | 0.68 | 0.72 | 0.65 | 0.68 |

The advantage of RF can be principally explained as follows. For the random forest classifier, node splitting is performed by calculating the relative importance scores of variables, ranking the variables, and selecting the variables with higher importance scores during the construction of a decision tree. Suppose there are n highly correlated variables; the importance scores of n − 1 variables are canceled out, ensuring that only one variable in this category is selected to participate in node splitting, thus avoiding information loss caused by the influence of highly correlated variables.

### 5.2. Accuracy

The brief results of the random forest model are shown in Tables 10–12. In summary, the average accuracies of the classifier on the training set, validation set, and test set are 0.84, 0.79, and 0.76, respectively. By comparing the F1-scores for the five lithofacies, the model achieves the best outcome on the training set for Lithofacies 2, with an F1-score of 0.88, while the model performance for Lithofacies 3 is better than that for all other facies on the validation set and test set, with F1-scores of 0.86 and 0.84, respectively.

**Table 10.** Performance summary of the training data.

| Facies | Precision | Recall | F1-Score | Support |
|---|---|---|---|---|
| 1 | 0.79 | 0.93 | 0.85 | 91 |
| 2 | 0.93 | 0.84 | 0.88 | 132 |
| 3 | 0.89 | 0.77 | 0.82 | 141 |
| 4 | 0.88 | 0.84 | 0.86 | 108 |
| 5 | 0.62 | 0.94 | 0.75 | 36 |
| Accuracy | 0.84 | | | 508 |

**Table 11.** Performance summary of the validation data.

| Facies | Precision | Recall | F1-Score | Support |
|---|---|---|---|---|
| 1 | 0.78 | 0.78 | 0.78 | 23 |
| 2 | 0.88 | 0.73 | 0.79 | 40 |
| 3 | 0.93 | 0.80 | 0.86 | 35 |
| 4 | 0.63 | 0.81 | 0.71 | 21 |
| 5 | 0.57 | 1.00 | 0.73 | 8 |
| Accuracy | 0.79 | | | 127 |

**Table 12.** Performance summary of the test data.

| Facies | Precision | Recall | F1-Score | Support |
|---|---|---|---|---|
| 1 | 0.53 | 0.74 | 0.62 | 72 |
| 2 | 0.86 | 0.76 | 0.81 | 233 |
| 3 | 0.89 | 0.80 | 0.84 | 267 |
| 4 | 0.52 | 0.69 | 0.59 | 104 |
| 5 | 0.88 | 0.75 | 0.81 | 51 |
| Accuracy | 0.76 | | | 727 |

Figure 17 displays the visual comparison between core-interpreted lithofacies and predicted lithofacies in wells FX184 and FY1. It is acceptable that the predicted lithofacies similarly duplicate the original lithofacies stacking pattern in both wells. The detailed model performances on the training set, validation set, and test set are illustrated in Figures 18–20, respectively.

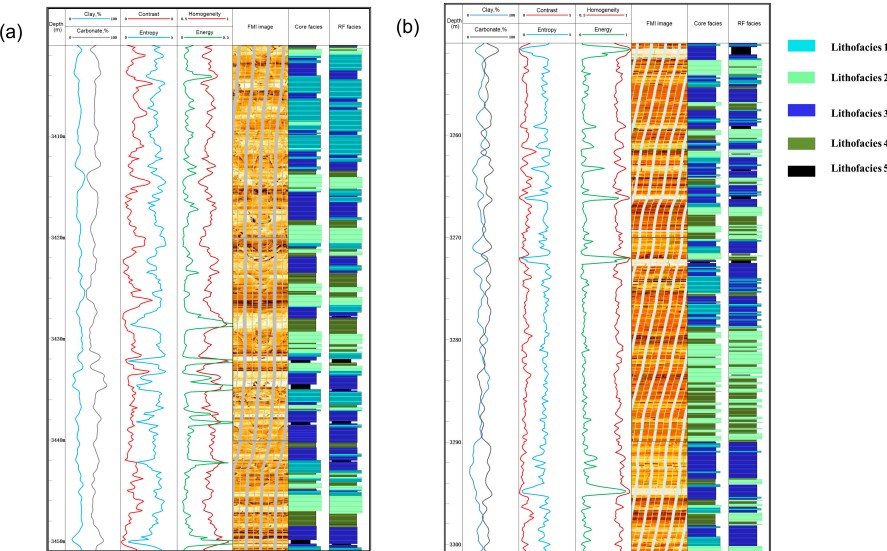

**Figure 17.** Comparison of core facies and RF facies in (**a**) well FX184 and (**b**) well FY1.

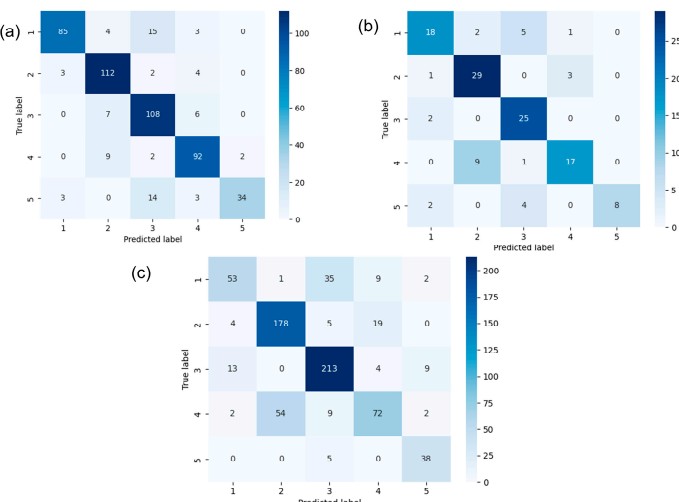

**Figure 18.** Unnormalized confusion matrices for the (**a**) training data, (**b**) validation data, and (**c**) test data.

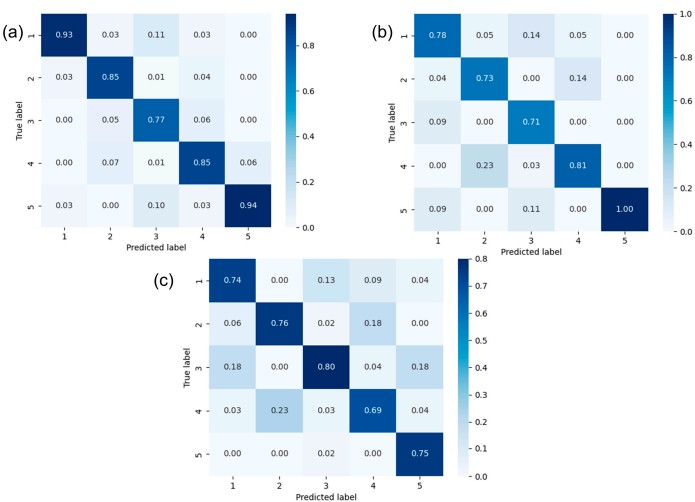

**Figure 19.** Precision matrices for the (**a**) training data, (**b**) validation data, and (**c**) test data.

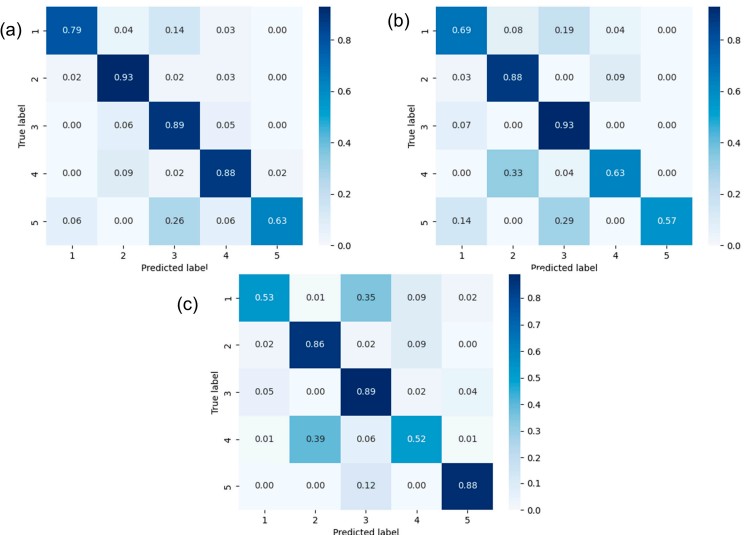

**Figure 20.** Recall matrices for the (**a**) training data, (**b**) validation data, and (**c**) test data.

## 6. Discussion

### 6.1. Misclassification Data Analysis

After comparing the core-interpreted lithofacies with the predicted lithofacies, it is found that the probability of lithofacies misclassification becomes higher in thin layers, i.e., layers whose thickness lies between 0.4 m and 1 m. This can be explained by the fact that the original lithofacies are manually determined by petroleum geologists, and frequent changes in lithofacies in thin layers may obscure the lithofacies interfaces. In addition, the prediction accuracy for well FY1 is not as good as that for well FX184. Test well FY1 is located in the lower part of the central Boxing Sag. The average clay mineral content of well FY1 is higher than that of well FX184, and more laminae are developed, which leads to the creation of thinner layers. Therefore, misclassification involving thin layers occurs more often in well FY1.

In addition, approximately 64% of the thin-layer misclassifications appear between Lithofacies 1 and Lithofacies 3. The reason lies in that Lithofacies 1 and Lithofacies 3 are very similar in terms of texture features and minerology logs, which presents a major challenge for the prediction model. It is also discovered that the classification accuracy of Lithofacies 5 is relatively lower than those of the other facies, which is primarily attributed to the small number of Lithofacies 5 samples. However, because the proportion of Lithofacies 5 in the total samples is relatively small, it does not exert a great adverse impact on the overall classification performance. In the future, with the increase in labeled lithofacies samples, the classification accuracy of thin layers and Lithofacies 5 can be improved.

### 6.2. Production Prediction

Using the GLCM-RF method, data from six wells in the Boxing Sag were subjected to lithofacies division, and the relationship between the thickness of the laminated lithofacies and daily oil production was analyzed. All six wells were pumped for production. As shown in Figure 21, the daily oil production exhibits a good linear relationship with the thickness of the laminated lithofacies, which confirms previous knowledge about laminated lithofacies. The organic-rich laminated lithofacies have various types of reservoir space and higher horizontal permeability, and the organic matter is distributed in a network. Therefore, it has better production capacity compared with the other lithofacies.

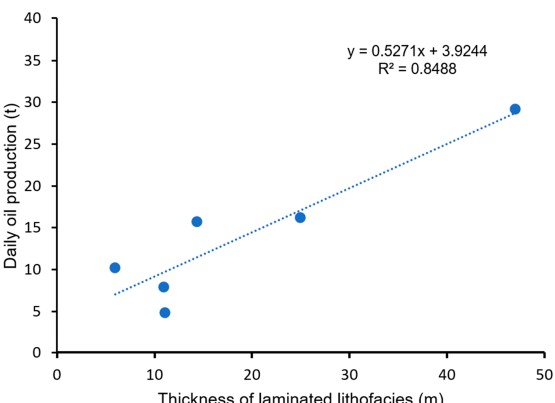

**Figure 21.** Relationship between the thickness of the laminated facies and daily oil production.

According to statistics, 70% of the shale oil delivery well sections in the Boxing Sag are identified as laminated lithofacies, of which laminated argillaceous limestone facies accounts for 37%, laminated calcareous mudstone facies accounts for 33%, layered argillaceous limestone facies accounts for 19%, and layered calcareous mudstone facies accounts for 9%.

## 7. Conclusions

The identification of shale lithofacies with different sedimentary structures is the key to commercial hydrocarbon production in the Boxing Sag. However, the present classification method based on conventional wireline logs cannot achieve the desired result. The aim of this study was to test a practical approach to identify lithofacies with an image feature extraction tool and a machine learning technique from advanced logs.

In the first section, lithofacies classification was carried out with the aid of integrated data, including core, FMI images, thin sections, and SEM images. Five shale lithofacies were classified based on sedimentary structures and mineral contents. In the target lower third and upper fourth members of the Shahejie Formation, the lithofacies change rapidly, and the vertical lithofacies combination is dominated by the interbed of laminated lithofacies and layered lithofacies.

In the second section, an approach integrating the GLCM and RF to classify lithofacies from FMI images and ECS logs was tested. The conclusions are as follows.

(1) The experiments show that the GLCM could be used to extract shale texture features. The shale laminae exhibit horizontal textures with thickness and density changes in the FMI images. The GLCM could characterize the texture efficiently and accurately based on the spatial distribution of the grayscale. After sensitivity analysis of extracted texture features from the GLCM, it was proven that four features, energy, homogeneity, contrast, and entropy, were more capable of identifying shale sedimentary structures.

(2) To address the strong correlation between the four texture features, a comparison between the RF and several other classifiers (KNN, NBC, SVM, and DT) showed that the RF has the advantage of achieving higher accuracy for correlated input variables both in principle and in practice. To further improve the predictive ability of the model, hyperparameter optimization of the RF model was conducted, and the average accuracies of this model on the training data, validation data, and test data were 0.84, 0.79, and 0.76, respectively. The blind well test demonstrated that the RF model was also applicable to uncored wells.

(3) The geostatistical inversion model established under the constraint of finely divided lithofacies could more delicately describe the distribution characteristics of lithofacies between wells to precisely predict the lithofacies between wells. On the basis of lithofacies division, it was preliminarily clarified that there was a good linear relationship between the thickness of laminated lithofacies and production capacity in shale reservoirs.

**Author Contributions:** Conceptualization, M.T. (Min Tian); methodology, M.T. (Min Tian); software, M.T. (Min Tian); validation, M.T. (Min Tian); formal analysis, M.T. (Min Tian); investigation, M.T. (Min Tian) and M.W.; resources, M.W.; writing—original draft preparation, M.T. (Min Tian); writing—review and editing, M.T. (Min Tian) and M.W.; supervision, M.T. (Maojin Tan); project administration, M.T. (Maojin Tan). All authors have read and agreed to the published version of the manuscript.

**Funding:** This research received no external funding.

**Data Availability Statement:** No new data were created or analyzed in this study. Data sharing is not applicable to this article.

**Conflicts of Interest:** The authors declare no conflict of interest.

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
