# Peer review of "Identification of Shale Lithofacies from FMI Images and ECS Logs Using Machine Learning with GLCM Features"

_processes, doi:10.3390/pr11102982_

Round 1

Reviewer 1 Report

The following comments may help improving the quality of the manuscript:

1. The quality of the figures should be improved (Figures 2, 14 and 18).

2. What do you mean by conventional well log suites (name the logs)? (Line 57)

3. It is suggested to summarize lines 69 to 105 in a table/tables.

4. Explain the depth calibration process (Line 158).

5. Is the percentage of any pore type known? (Line 179)

Reviewer 2 Report

I have some questions and suggestions that I hope the authors can consider:

1. The article mentions conventional wireline logging several times. It is recommended to provide a brief introduction to what conventional wireline logging encompasses;

2. In line 20, the author mentions, "It can be concluded that at present, image feature extraction methods are mostly used to classify lithologies from FMI images, with few studies on the division of sedimentary structures and even fewer applications in shale." However, the references regarding this content are all from before 2014. Can they represent the current research status? Please add some new references.

3. In section 3.2 "lithofacies classification," what is the vertical resolution of the FMI images used? This section also mentions significant differences in mineral content for different lithofacies. It is recommended to include ECS log curves alongside the FMI images to more intuitively reflect mineral content variations and also specify the vertical resolution of the ECS logging;

4. What is the physical size of the 230-pixel-length FMI images used in the article?

5. How thick are the lithofacies defined in the article? In Figure 11, is it a single lithofacies category obtained from an FMI image or are lithofacies determined at each depth?

6. The term "FM static image" appears many times in the article. Is there a phrasing error?

7. The article mentions the use of clay and carbonate rock minerals obtained from ECS logging, but in Figure 11's RF input image, there are three types of minerals in the ECS log image;

8. Figure 3's well log curve lacks units for depth. The image quality of the figures in the paper is low, it is recommended to improve the image resolution;

9. Section 6.2.1 appears to have weak relevance to the content of the paper. It is suggested to remove it.

Please consider these points and make appropriate revisions to the article.

None

Reviewer 3 Report

This study proposes a precise shale lithofacies classification method using the GLCM and RF algorithms based on FMI and ECS logging. GLCM extracts texture features from FMI images, and RF handles classification, achieving high accuracy (0.76-0.84). Application in lithofacies and production prediction is also discussed. However, before further consideration of the manuscript, the authors must “fully” address the comments listed below:

1.      Could you explain in detail how GLCM is used to extract texture features from FMI images and how these features reflect shale sedimentary structures?

2.      Can you provide a more comprehensive explanation of the random forest algorithm's suitability for supervised lithofacies classification, especially in handling correlated input features?

3.      How do the carbonate content and clay content calculated from ECS logs contribute to improving the model's ability to differentiate between argillaceous limestone and calcareous mudstone in lithofacies classification?

4.      Can you provide a detailed geological description of the Boxing Sag and its geological formations, including their significance in the study?

5.      The authors mentioned that “Among them, n_estimators can significantly impact the overall accuracy of the model”. However, this statement is partially true because the dataset might be highly complex such that even a better/more optimized network may not necessarily improve the model accuracy. In machine learning, this can refer to “Kolmogorov complexity” denoting the length of the shortest computer program that produces the output. Therefore, please write a paragraph in your paper arguing that reducing the complexity of your dataset can potentially improve the accuracy of the deep learning model (you may read and reference the two journal papers that efficiently leveraged decreased dataset complexity to rapidly improve their model accuracy: paper 1: https://doi.org/10.1007/s10462-019-09750-3, and paper 2: https://doi.org/10.1038/s41598-023-28763-1)

6.      What challenges are associated with distinguishing shale sedimentary structures from conventional wireline logs, and how do the FMI logs and ECS logs overcome these challenges?

7.      Can you provide a more detailed description of the five defined lithofacies, including their mineral compositions, pore types, and organic matter content, and how they were differentiated in the study?

Round 2

Reviewer 3 Report

The authors addressed my comments.